# Impact of IRS: Four-years of entomological surveillance of the Indian Visceral Leishmaniases elimination programme

Rinki Deb[1], Rudra Pratap Singh[1], Prabhas Kumar Mishra[2], Lisa Hitchins[1], Emma Reid[1], Arti Manorama Barwa[2], Debanjan Patra[2], Chandrima Das[2], Indranil Sukla[2], Ashish Kumar Srivastava[2], Shilpa Raj[2], Swikruti Mishra[2], Madhuri Swain[2], Swapna Mondal[2], Udita Mandal[2], Geraldine M. Foster[1], Anna Trett[1], Gala Garrod[1], Laura McKenzie[1], Asgar Ali[2], Karthick Morchan[2], Indrajit Chaudhuri[2], Nupur Roy[3], Naresh K. Gill[3], Chandramani Singh[4], Neeraj Agarwal[4], Sadhana Sharma[4], Michelle C. Stanton[1], Janet Hemingway[1], Sridhar Srikantiah[2], Michael Coleman[1]*

1 Liverpool School of Tropical Medicine, Liverpool, United Kingdom, 2 CARE India, Patna, India, 3 National Vector Borne Disease Control Programme, Directorate General of Health Services, Ministry of Health and Family Welfare, Delhi, India, 4 All India Institute of Medical Sciences, Patna, India

* Michael.Coleman@lstmed.ac.uk

**Data Availability Statement:** Data is contained in the manuscript and supporting files.

**Funding:** This work was funded by the BMGF grants, https://www.gatesfoundation.org,

## Abstract

### Background

In 2005, Bangladesh, India and Nepal agreed to eliminate visceral leishmaniasis (VL) as a public health problem. The approach to this was through improved case detection and treatment, and controlling transmission by the sand fly vector *Phlebotomus argentipes*, with indoor residual spraying (IRS) of insecticide. Initially, India applied DDT with stirrup pumps for IRS, however, this did not reduce transmission. After 2015 onwards, the pyrethroid alpha-cypermethrin was applied with compression pumps, and entomological surveillance was initiated in 2016.

### Methods

Eight sentinel sites were established in the Indian states of Bihar, Jharkhand and West Bengal. IRS coverage was monitored by household survey, quality of insecticide application was measured by HPLC, presence and abundance of the VL vector was monitored by CDC light traps, insecticide resistance was measured with WHO diagnostic assays and case incidence was determined from the VL case register KAMIS.

### Results

Complete treatment of houses with IRS increased across all sites from 57% in 2016 to 70% of houses in 2019, rising to >80% if partial house IRS coverage is included (except West Bengal). The quality of insecticide application has improved compared to previous studies, average doses of insecticide on filters papers ranged from 1.52 times the target dose of 25mg/m$^2$ alpha-cypermethrin in 2019 to 1.67 times in 2018. Resistance to DDT has continued to increase, but the vector was not resistant to carbamates, organophosphates or

OPP1151797 to MC and OPP1196454 to SS. The funders had no role in study design, data collection and analysis, decision to publish, or preparation of the manuscript.

**Competing interests:** The authors have declared that no competing interests exist.

pyrethroids. The annual and seasonal abundance of *P. argentipes* declined between 2016 to 2019 with an overall infection rate of 0.03%. This was associated with a decline in VL incidence for the blocks represented by the sentinel sites from 1.16 per 10,000 population in 2016 to 0.51 per 10,000 in 2019.

## Conclusion

Through effective case detection and management reducing the infection reservoirs for *P. argentipes* in the human population combined with IRS keeping *P. argentipes* abundance and infectivity low has reduced VL transmission. This combination of effective case management and vector control has now brought India within reach of the VL elimination targets.

## Author summary

Visceral Leishmaniasis (VL), also known as kala-azar, is a major parasitic disease in South Asia (Indian subcontinent), with 85% of the disease incidence in India. Historically VL had been controlled and almost eliminated with Indoor Residual Spraying (IRS) using dichlorodiphenyltrichloroethane (DDT). However, reinitiating this approach in 2015 failed due to high insecticide resistance in the sand fly vector and poor IRS quality, meaning that VL elimination targets were not met. To improve this the National Vector Borne Disease Control Programme changed to an effective insecticide, alpha-cypermethrin and altered the mode of application to compression pumps. Sentinel sites were established to monitor the entomological indicators, these showed the positive impact of these changes from 2016 to 2019. During this period the overall incidence of disease has decreased, and India is now on track to reach it's target incidence for VL of less than 1/1000 people at the sub-district (block) level.

## Introduction

Indoor residual spraying (IRS) involves the application of insecticide formulations to the interior walls of houses, animal shelters and public buildings where people are at risk of transmission of insect borne diseases. The organochlorine insecticide, dichlorodiphenyltrichloroethane (DDT) was introduced for vector control in 1946 [1], before being used at scale during the Global Malaria Eradication Campaign between 1955–1969 [2–4] resulting in the elimination of malaria from 37 countries.

Visceral leishmaniasis (VL) is caused by the parasite *Leishmania donovani* and in South Asia (Indian subcontinent) is transmitted by the bite of the female sand fly *Phlebotomus argentipes* [5,6]. From 1953–1962 DDT-based IRS was carried out in India by the national malaria control programme. This had the secondary impact of controlling *P. argentipes* and almost eliminating VL [7]. After IRS for malaria ceased in VL endemic areas, DDT IRS was used intermittently to control VL outbreaks between 1977–1979 and 1992–1995 [8,9]. Between 2004–2010 there were an estimated 200,000–400,000 new cases of VL annually [10], with 67% of these occurring in Bangladesh, India and Nepal. Today in India, 130 million people from 54 districts within the four endemic States of Bihar, Jharkhand, Uttar Pradesh and West Bengal remain at risk of VL. The World Health Organization (WHO) still estimates 300,000 annual

cases of VL globally [11], however, only 16,970 cases were recorded in 2018 in the Global Health Observatory data repository [12], reflecting enhanced VL control measures.

In 2005, a tripartite agreement between Bangladesh, India, and Nepal was signed with the aim of eliminating VL and post–kala-azar dermal leishmaniasis as a public health problem *i.e.* to less than one case per 10,000 population by 2015 [13]. Elimination was attempted using a combination of vector control, rapid diagnosis and treatment of the disease [6,14].

The VL elimination programme was planned in four phases: the preparatory phase which involved initiating improved case detection and biannual IRS; the attack phase where prevention and treatment activities were scaled up and monitoring was increased; the consolidation phase when the elimination target should be reached, and post elimination validation to maintain elimination, when surveillance is scaled-up to avoid resurgence [15]. India aimed to reach the consolidation phase by 2015, but timescales were revised to 2017 and then 2020. Despite the disruption of the Coronavirus (COVID-19) global pandemic in 2020, IRS vector control activities have been maintained to reach the elimination target.

In elimination settings, an integrated vector management approach is ideal [13], but there are limited data to demonstrate the impact of different vector control methods on VL transmission. A cluster randomized trial in Bangladesh, India, and Nepal demonstrated that insecticide-based IRS reduced the indoor abundance of *P. argentipes* by 72.4% in intervention clusters compared with controls; this effect was greater than the effect of environmental modification (42% reduction) or the use of long-lasting insecticide-treated nets (43.7% reduction) [16]. Transmission models also suggest that IRS is capable of achieving VL elimination if sand fly abundance can be reduced by 67% [17]. Hence, if case detection and treatment with effective drugs and effective IRS are combined, elimination of VL as a public health problem should be feasible [6,14].

In India, to maximize the impact IRS is carried out in houses and cattle shelters as *P. argentipes* shows endophilic and exophagic behaviour [18]. Cattle sheds are included as *P. argentipes* collected outdoors and in cattle sheds using CDC light traps [19] have predominantly fed on humans [19,20]. Based on previous success, India initially used DDT IRS, applying a wettable powder formulation at 1g/ m$^2$ with stirrup pumps. However, by 2013, progress towards the consolidation phase was limited [13]. Operationally relevant levels of resistance to DDT in *P. argentipes* and sub-optimal dose delivery of IRS were identified as key barriers to success [21]. This prompted the National Vector Borne Disease Control Programme (NVBDCP) to switch to a pyrethroid insecticide, alpha-cypermethrin 5% wettable powder, to overcome resistance [22] and to compression pumps (Hudson X-pert Sprayer) to improve the quality of IRS delivery, in accordance with WHO guidelines for IRS [23].

To monitor the impact of vector control, the systematic entomological tracking of vector species and their characteristics is critical [24]. WHO defines entomological surveillance as the regular, systematic collection, analysis and interpretation of entomological data for risk assessment, planning, implementation, monitoring and evaluation of vector control interventions with key indicators including the abundance of the vector species and insecticide resistance [25,26]. In India, this was undertaken in collaboration with the NVBDCP using routine sentinel surveillance from 2016 onwards; to assess the impact on disease burden, VL case incidence was tracked along with entomological indicators.

## Methods

### Ethics statement

Ethics for this study were obtained from: All India Institute of Medical Sciences Patna, Institute Ethics Committee, Ref. No. AIIMS/Pat/IEC/2019/412; Rajendra Memorial Research

Institute of Medical Sciences Indian Council of Medical Research Ethics Committee, Ref. No. 13/IEC/2015; Ashirwad Ethics Committee, Ashirwad Hospital and Research Centre Ref. No. CARE/INDIA/Jan/20/01 and Liverpool School of Tropical Medicine, LSTM Research Ethics Committee, Ref. No. 15.023. Written consent was obtained from all households for the collection of sand flies and verbal consent was obtained for the IRS household questionnaire in the study.

## Sentinel sites

Eight sentinel sites in VL endemic areas were established: six in Bihar, one in Jharkhand and one in West Bengal. Each site had at least 1 new VL case per 10,000 persons per year at sub-district (block) level. In the State of Bihar, the sites also represented ecologically diverse regions. Block selection was based on total reported VL case numbers, extracted from the 2015 district level IRS micro plan data. At the village level, criteria for selection included: VL case history for the previous three consecutive years, appropriate infrastructure to allow year-round village access and absence of additional planned field research activities. Of the villages that met the selection criteria, four IRS villages per sentinel site were selected using a random number generator in Microsoft Excel. A further two villages with no history of IRS or VL cases for the previous five years were also selected per sentinel site using the same random number generator method, to monitor any social or seasonal effects on entomological indicators that are unrelated to IRS.

## Indoor residual spray routine coverage data

Routine coverage data by IRS round for 2016–2019 for the sentinel site villages were obtained from spray supervisor registers, held at the District Malaria Office. Where possible, data on the rooms, cattle sheds and verandas targeted for treatment and whether these were sprayed, locked or refused were digitised. Coverage rates (sprayed, locked and refused) were calculated for rooms and cattle sheds.

## Indoor residual spray survey coverage data

IRS coverage data was obtained from community-based cross-sectional studies conducted biannually between March—June and July—September, in the VL sentinel districts of Bihar and Jharkhand (West Bengal was not included in this survey). A sample size of 800 households per district were targeted for each spray round in each of these districts assuming a 95% confidence level, 5% absolute precision and 50% expected coverage based on the most conservative measure and accounting for any cluster effects.

In each district, 40 villages were selected from the operational IRS plans using Probability Proportional to Size. From each village 20 houses were systematically selected with a random start (The Index/Starting point was selected randomly from Anganwadi's household survey register, where each house was numerated, using a random number table). Interviews were carried out manually or using Computer Assisted Personal Interview (CAPI) tools in the local language to assess if houses and cattle sheds had been completely sprayed, partially sprayed or not sprayed. The sampling was designed to provide estimates of IRS coverage for the villages targeted for spraying in each IRS round in each district. Surveys were typically completed within a month of completion of each round of IRS.

## Quality assurance

Samples for quality assurance were collected from all 8 sentinel sites from 2017 to 2019. Surveys were conducted in rooms within houses where *P. argentipes* abundance monitoring was ongoing.

To determine the concentration of alpha-cypermethrin delivered to walls during IRS, $5cm^2$ Whatman Grade 1 filter papers were affixed onto all four walls of the room prior to IRS, as described by WHO [26]. Single filter papers were affixed between 2–4 ft from the ground, on all four walls within the bedroom and stored at -4˚C until analysis following IRS activities.

The concentration of insecticide present on the filter papers was determined using high performance liquid chromatography (HPLC). All filter papers were cut into pieces of $\sim 1\ cm^2$. Five ml of a heptane/1-propoanol mixture (9:1) containing 100 μg of the internal standard dicyclohexyl phthalate (DCP) was added, and samples were sonicated for 15 min to extract the alpha-cypermethrinin. The insecticide extract (1ml) was transferred to a clean glass tube and evaporated to dryness at 60˚C. One ml of acetonitrile was added, and the mixture was vortexed for 1 min to mix.

HPLC analysis was performed by injection of 20-μL aliquots of extract on a reverse-phase Hypersil GOLD C18 column (75 Å, 250 × 4.6 mm, 5-μm particle size; Thermo Scientific) at 23–25˚C. A mobile phase of acetonitrile/water (70:30) was used at a flow rate of 1 mL·min−1 to separate alpha-cypermethrin and DCP. The quantities of alpha-cypermethrin and DCP were calculated from standard curves established by known concentrations of authenticated standards. Peaks were detected at 232 nm with the Agilent 1260 Infinity Quaternary LC system, model G1311C detector (Agilent) and were analysed with Agilent OpenLAB CDS software.

Final alpha-cypermethrin content in grams per square meter was estimated using the following equations:

$$B = (P/V)xDx(100/E)xC$$

Where alpha-cypermethrin ($g/m^2$) (A) = ((B/S)x10,000)/1,000,000, B = alpha-cypermethrin (μg/25cm), P = peak area, V = slope value, D = dilution factor (5), E = extraction efficiency (100%), C = DCP correction factor, S = surface area of filter paper ($25cm^2$).

HPLC results were compared with the intended IRS target alpha-cypermethrin concentration on the wall of $25.0mg/m^2$. A 20% cut-off threshold was used to classify results whereby a concentration of less than $20.0\ mg/m^2$ was considered an under-spray, a range of 20.0–30.0 $mg/m^2$ was considered within the target range, and a concentration of greater than $30mg/m^2$ was considered an overspray.

## Insecticide susceptibility assays

Female *P. argentipes* were collected using mouth aspirators inside houses, verandas, and cattle sheds. Collected sand flies were exposed to alpha-cypermethrin (0.05%, 0.065% and 1%), bendiocarb (0.1%), deltamethrin (0.05%), DDT (4%) or malathion (5%) using WHO-impregnated filter papers following the WHO susceptibility test procedures [27]. Mortality was recorded after 24 hours. Controls were performed for each test using appropriate papers, and Abbott's formula was applied where necessary [28].

## Phlebotomus argentipes abundance

Year-round *P. argentipes* abundance was monitored using CDC light traps operating in 15 randomly selected houses in each village over a period of two consecutive nights (6:00 PM to 6:00 AM) on a bi-monthly basis. The light traps were hung in the corner of a bedroom and optimally positioned 15 cm away from the wall and 5 cm above ground. All sand flies were identified to species level by morphological criteria from established taxonomic keys [29].

Abundance was analysed by a generalised additive model (GAM) being fitted to the data aggregated to the village-level and monthly time scale to model the changes in sandfly

abundance over time, accounting for effect of IRS status (IRS or Non-IRS) of the village. Seasonal effects were modelled as a cubic regression spline, whereas long-term temporal trends were modelled using thin plate regression splines. A first order autoregressive component (AR (1)) was included in the model to account for temporal correlation.

### Identification of L. donovani in P. argentipes

*L. donovani* parasite kinetoplastid DNA (kDNA) in *P. argentipes* sand flies was detected by RTPCR [30]. Analysis was initially done on pooled sand fly DNA (maximum of 8 sand flies per pool), any positive pooled results were investigated further at the individual sand fly level.

### Case data for sentinel sites

Total annual case data (2016 to 2019) for the blocks containing the sentinel sites was extracted from the Kala-Azar Management Information System (KAMIS) database. Population calculated using Government of India 2011 Census [31] and population projected using the formula: Population Final = Population initial ((1+(growth rate/100))^(time in years)). In the absence of open-access data on the population at risk of VL within Bihar, the total state population was used as the denominator to calculate incidence per 10,000 [7]. Case incidence rate per 10,000 = (total number new cases/persons at risk) x10,000.

## Results

### Sentinel sites

The eight VL sentinel sites were established in a phased approach starting with Muzaffarpur and Samastipur in April 2016 and ending with Darjeeling in November 2017 (S1 Table). Fig 1 shows the location of the sentinel sites (Darjeeling, East Champaran, Godda, Gopalganj, Katihar, Muzaffarpur, Purnia and Samastipur). Within each sentinel site, a total of four IRS villages and 2 non-IRS villages were monitored.

Where possible, this ratio of non-IRS to IRS villages was maintained. However, over the three years of monitoring, four changes in IRS status occurred in response to the emergence of VL cases in non-IRS villages (S1 Table). In Purnia, both non-IRS villages were sprayed within 12 months of collections starting, and in 2019, after a spike in VL cases in Gopalganj, both non-IRS villages were sprayed. Randomly selected houses within the sentinel sites remained fixed to enable longitudinal monitoring unless house owners opted to leave the study. These houses were replaced with houses selected using the same random selection methodology.

### Indoor residual spraying data

In order to be effective IRS needs to be applied at the optimal time of year, with high coverage of all targeted structures at an accurate dosage [32]. In the current programme, biannual spraying is targeted in March and August, with spray operators reporting over 80% coverage of households and cattle sheds covered with alpha-cypermethrin at 25mg/m$^2$ IRS.

### IRS coverage as reported by spray teams

IRS coverage data was provided by NVBDCP at the spray team level from 2017–2019 for each of the sentinel sites (Fig 2). Due to operational issues only one round of IRS was undertaken in 2018. Data obtained from the spray registers completed by spray teams predominantly showed high levels of IRS coverage for household rooms (79.3–99.7%) and cattle sheds (68.3–100%) across all spray rounds (S2 Table).

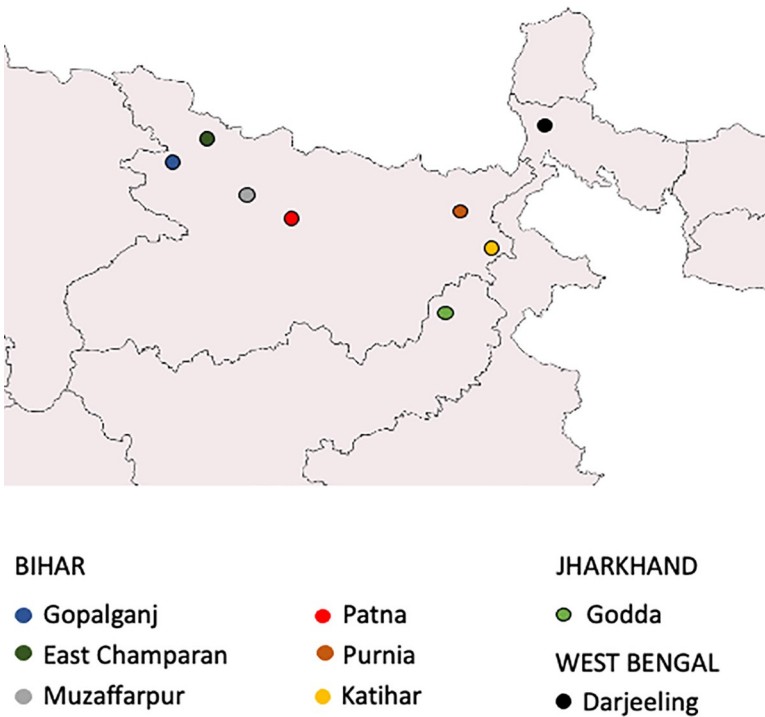

**Fig 1. Map of the three VL endemic areas in India with the sentinel site districts labelled.** (Base layers for map can be found at https://www.statsilk.com/maps/download-free-shapefile-maps).

## IRS coverage as measured from survey data

The household surveys indicated much lower complete house IRS coverage values (28 to 79%) than self-reported by the spray teams. The IRS coverage survey data for the sentinel sites located in Bihar and Jharkhand from 2016–2019 indicates that the WHO recommended minimum 80% coverage complete coverage for successful IRS [23] was not met across all sentinel sites, although the 80% target was achieved if household bedroom coverage data, where most transmission occurs is considered. In 2016, coverage for complete spray ranged from 28% in Godda, to 77% in Katihar. By 2019, an improvement in IRS coverage was seen, with the minimum coverage (complete spray) achieved in in Samastipur (54%) and the maximum coverage in Purnia (77%). In Bihar, consistently low-level of complete coverage over the four years was observed in Samastipur (49–61%). The greatest improvement in spray complete IRS coverage was observed in Godda, Jharkhand where in 2016, 28% complete coverage was reported, which increased to 75% in 2019, Fig 2. The 2019 household surveys reported a much higher complete coverage (54–77%) suggesting an overall improvement in the spray programme that can be seen in for each sentinel site in Fig 2. However, if partial spray coverage is included the percentage coverage ranges from 74% to 93% (S3 Table).

## Quality assurance (QA)

A total of 642 houses that received IRS across the eight sites were included in the QA surveys. Four filter papers per wall were affixed in the bedroom prior to IRS and recovered afterwards. A total of 2,992 Whatman filter papers were retrieved and analysed by HPLC over three years (2017–2019). In 2017, all 2,140 filter papers collected from field surveys were analysed. In subsequent years a minimum random sample of 20% of houses from each sentinel site were

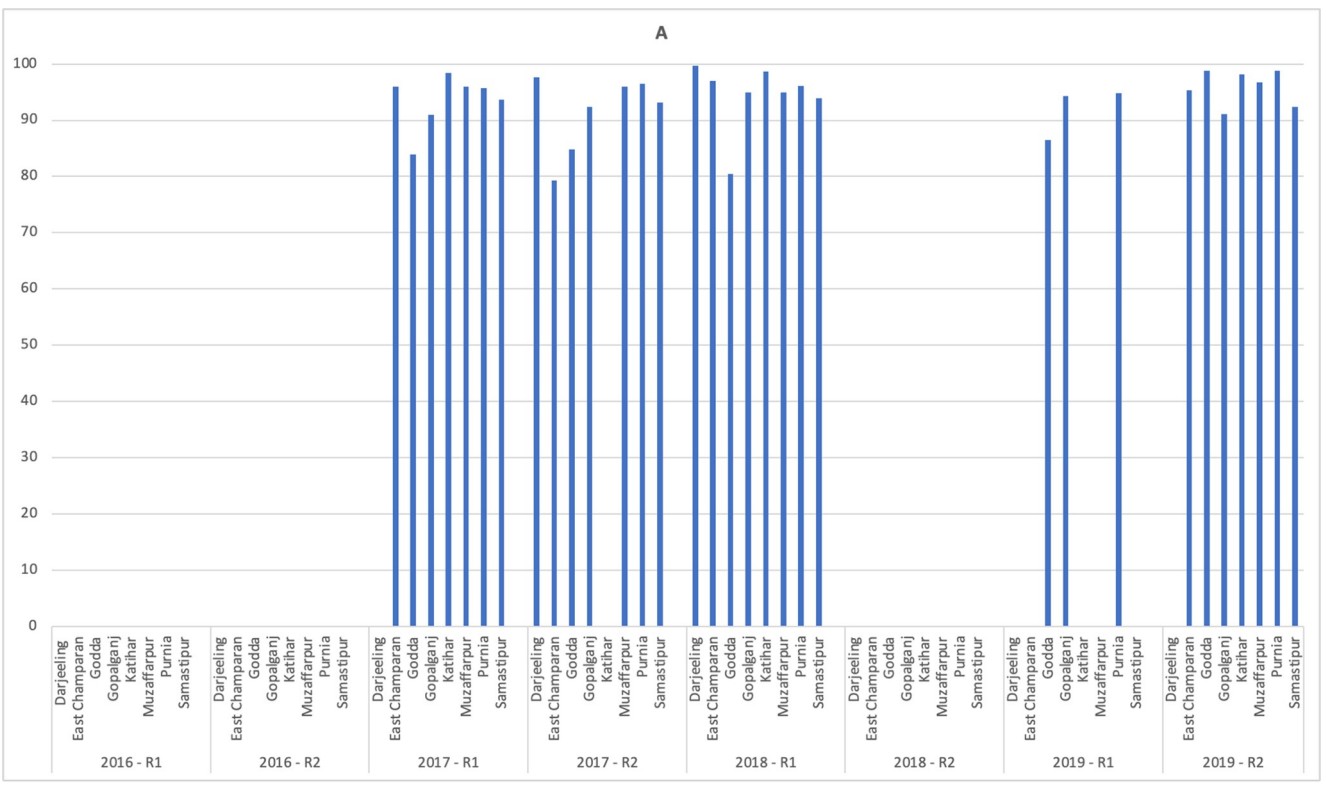

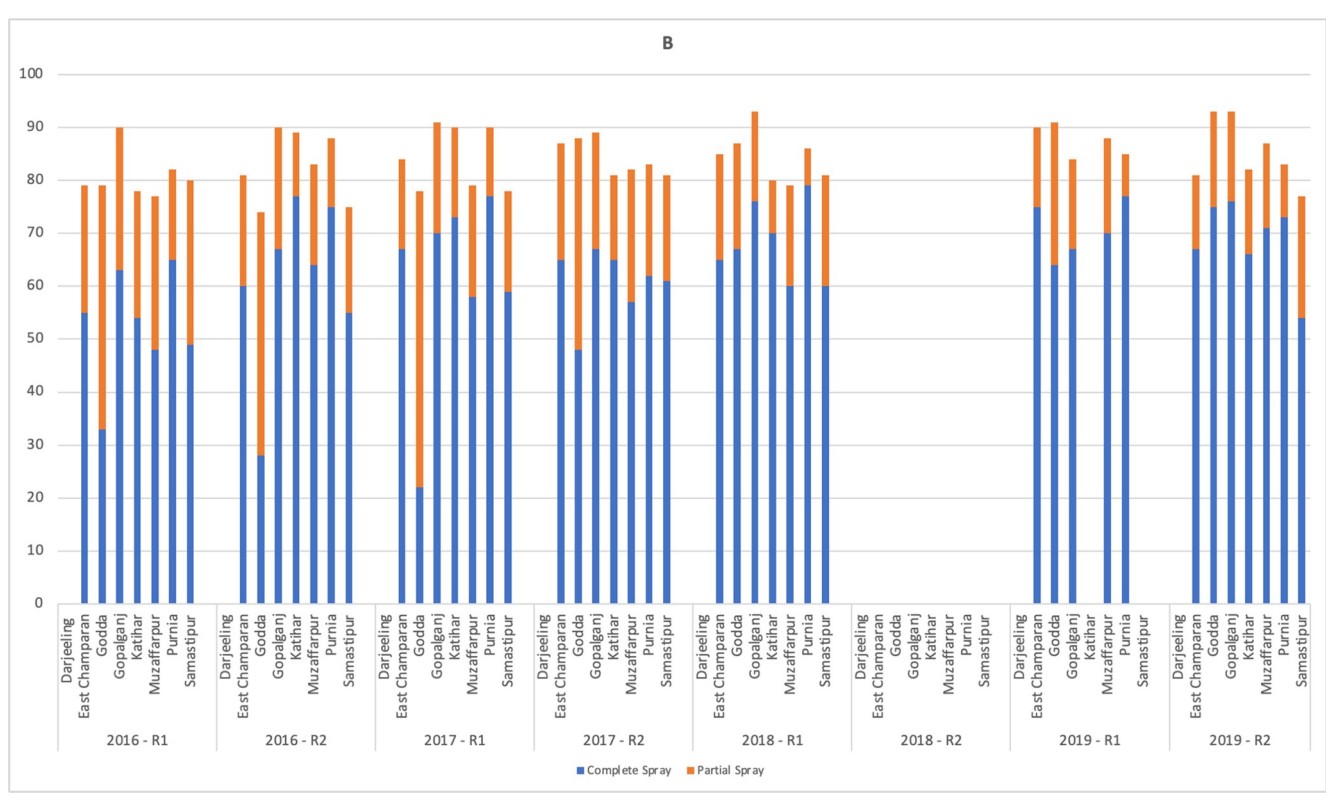

**Fig 2. Percentage of household structures that were completely sprayed in each IRS round as reported by; A spray operators in all 8 sites, and B sentinel sites surveys.** R1 = Round one R2 = Round two. Solid colours represent complete spray and lined colours partial spray.

analysed. The total number of filter papers retrieved per year varied due to change in IRS status of some of the villages, or programmatic decisions to spray villages where there was no VL history. In addition, upon retrieval some filter papers were found to be missing from the affixed position and therefore no sample was available.

Of the 2,992 filter papers tested from 2017 to 2019 only 15.4% had been sprayed at the target dose ($25mg^2 \pm 10\%$ range 20-30mg/m$^2$). The best IRS quality was observed in Samastipur with 25.71% of filter papers on target during round one of IRS in 2018 (Fig 3). High levels of under spraying (86.67%) were observed in Katihar during 2017's first round of IRS: spray performance improved in subsequent years with 58.33% of filter papers analysed under sprayed in 2019 round 2. Consistently high levels of over-spraying were observed in Gopalganj for all three years of filter paper analysis (55.56–91.67%). In East Champaran the quality of spraying declined annually from 48.3% under spray in round 1 of 2017 to 75% under spray in round 2 of 2019.

Considering all the filter papers collected from 2017–2019 the average concentration of insecticide by year, irrespective of geography was relatively consistent over the time period (2017: 40.03mg/m$^2$, 2018: 41.64mg/m$^2$ and 37.92mg/m$^2$). The average doses on the filter papers were 1.6 times higher than the target concentration of insecticide (25mg/m$^2$) in 2018 this was 1.67 times and 2019 1.52 times. This overdosing may in part be due to IRS operators over spraying the filter papers. The highest level of over spray was detected in Muzaffarpur in 2018 (657.1mg/m$^2$). During the 2018 IRS campaign the target dose range (20-30mg/m$^2$) was achieved in surveys from three of the eight districts.

## Susceptibility assays

The wild caught female *P. argentipes* mortality ranged from 39.9–66.7% for 4% DDT after exposure to WHO insecticide impregnated papers. Minimal resistance was detected to alpha-cypermethrin at any of the three concentrations tested (0.05%, 0.0675% and 1%) with mortality ranging from 97.6–100% during this period. No resistance was detected to the other insecticides tested, Table 1.

## Phlebotomus argentipes abundance

Over the three and a half years a total of 102,951 CDC light trap collections were performed in the eight sentinel sites, from which a total of 91,571 female *P. argentipes* sand flies were

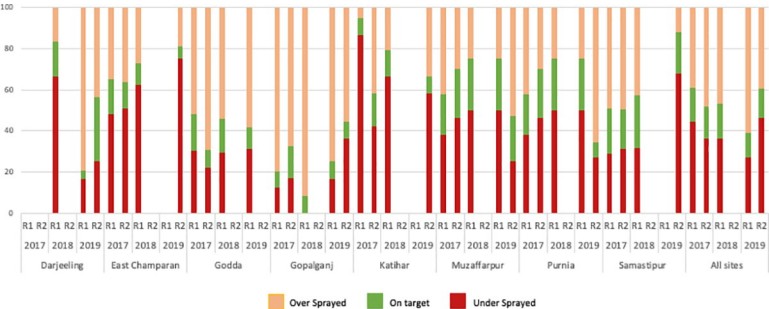

**Fig 3. Percentage of household structures that received the correct dose, overdose and under dose of alpha-cypermethrin.** R1-Round 1; R2-Round 2; % over sprayed- >30 mg/m$^2$; % on target-20-30mg/m$^2$; % under sprayed —<20 mg/m$^2$ alpha-cypermethrin.

**Table 1. *P. argentipes* mortality over time for a range of insecticides in WHO susceptibility tests.**

| Insecticide | 2016 | | 2017 | | 2018 | | 2019 | |
|---|---|---|---|---|---|---|---|---|
| | %mort | n | %mort | N | %mort | N | %mort | N |
| alpha-cypermethrin 0.05% | 99.62 | 527 | 97.60 | 1126 | 99.87 | 4566 | 99.47 | 2340 |
| Alpha-cypermethrin 0.0675% | 97.87 | 947 | 99.01 | 1997 | 99.49 | 4219 | 98.68 | 2424 |
| Alpha-cypermethrin 0.1% | 100 | 214 | 97.99 | 1280 | 99.71 | 3236 | 99.74 | 2920 |
| Bendiocarb 0.1% | | | 98.73 | 1007 | 98.49 | 3063 | 100 | 745 |
| Deltamethrin 0.05% | | | 99.63 | 1142 | 98.93 | 2969 | 99.93 | 1597 |
| DDT 4% | 66.67 | 63 | 49.55 | 1351 | 44.35 | 2316 | 39.90 | 1377 |
| Malathion 5% | | | 98.86 | 1100 | 99.96 | 2737 | 100 | 905 |

identified. In IRS villages, a total of 62,384 (69,450 collections) *P. argentipes* sand flies were collected. In comparison a total of 29,187 (33,501 collections) *P. argentipes* sand flies were collected from non-IRS villages. The peak period for sand fly abundance irrespective of village IRS status was between June and September with a peak abundance reaching 2.92 (July-2016), 3.00 (July-2017), 1.81 (July-2018), 1.26 (September-2019) sand flies per trap per night in IRS villages and 3.38 (July-2016), 2.52 (July-2017), 2.05 (June-2018) and 1.77 (July-2019) sand flies per trap per night in non-IRS villages.

Throughout the study period a general annual decline in *P. argentipes* abundance was observed (log relative risk = -0.01684, P = 0.0542, 95% CI 0.9668–1.0000) (Table 2). Previously abundance has been reported at 4.0 to 5.5 *P. argentipes*/trap/night for 2014, in IRS villages [21], and here we report a reduction to 0.75 *P. argentipes*/trap/night by 2019.

Fig 4 shows the monthly trends in *P. argentipes* abundance aggregated by village-level IRS status. On fitting a GAM to the village-level monthly abundance data, it was noted that the smoothed long-term temporal trend was very close to linear, therefore time was included as a linear term rather than a smoothed function in the model. In the resulting model both the long-term temporal trend and the smoothed seasonal trend (s(month)) were significant, whereas IRS status was not significant (p = 0.5086). This indicates that after accounting for seasonal trends (Fig 4), no discernible difference in trends observed in IRS and non-IRS villages (p = 0.5679). This along with the GAM analysis suggests that there was no seasonal or social activity, e.g., lime plastering of walls impacting IRS [26].

## Identification of L. donovani in P. argentipes

A total of 14,775 *P. argentipes* were assessed for the presence of *L. donovani* across all IRS sentinel site villages. Only four sand flies from East Champaran were positive (Table 3), suggesting that there is a very low active transmission of VL at the sentinel sites.

## Case data for sentinel sites

From 2016 to 2019 a total of 764 VL cases were reported in the blocks with the sentinel sites. A steady decline in incidence was observed in all blocks, apart from Darjeeling, Gopalganj and

**Table 2. Generalised Additive Model Analysis result of *P. argentipes* abundance data from April 2016 to December 2019.**

| GAM Analysis | | | |
|---|---|---|---|
| Parameter | log-RR | 95% CI | P-value |
| Month (Linear) | -0.0168 | 0.9668–1.0000 | 0.0542 |
| IRS | -0.0036 | 0.8194–1.4392 | 0.5086 |

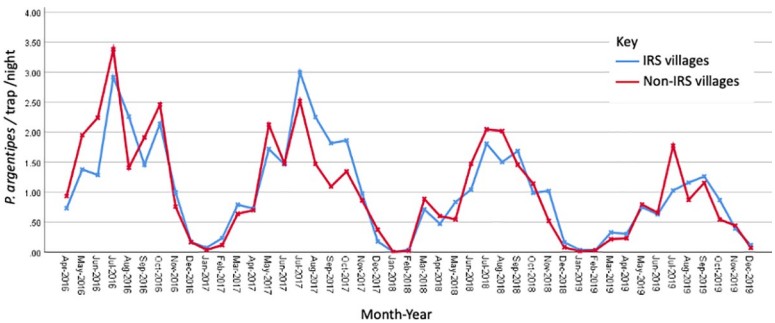

**Fig 4. Aggregated *P. argentipes* abundance collected in all sentinel sites for IRS and non-IRS villages.**

Samastipur, where an increase was seen in 2018. The highest incidence of 3.63 cases per 10,000 was observed in 2018 in Godda which along with Gopalganj consistently had high case numbers. Other than Gopalganj, the threshold for VL elimination of an incidence less than 1/1000 people at the block level, was reached and sustained from 2018 onwards in the sentinel site blocks, with the lowest incidence of 0.042 being observed in Samastipur in 2019, shown in Table 4. When aggregated there is a decline in incidence from 1.16 VL cases per 10,000 in 2016 to 0.51 VL cases per 10,000 in 2019, which is below the elimination target.

## Limitations

As the surveillance data collection presented here was initiated 10-years after the beginning of the VL elimination programme there is no true-baseline available for comparison. This is also an observational study based on the 8 VL surveillance sites for data collection, which represents only 8 of the 600 blocks that are endemic for VL. While analysis of trends has been limited to these sites, the assumption of reduced incidence across all blocks is due to the overall impact of enhanced case detection, treatment and vector control.

## Discussion

VL elimination as a public health problem is defined as reducing the annual incidence to <1 case per 10,000 people at the block level [6]. In 2002 the National Health Policy of the Government of India was to eliminate VL from the region by 2010 [4,5]. In 2005, the governments of Bangladesh, India, and Nepal developed a strategic regional framework to eliminate VL as a public health problem by 2015 [4,5]. This original target assumed that a VL vaccine in late-

**Table 3. Number of *P. argentipes* with positive *L. donovani* detection at each sentinel site by year.**

|  | 2017 | | 2018 | | 2019 | | Total | |
|---|---|---|---|---|---|---|---|---|
|  | **n. Tested** | **n. +ve** | **n. Tested** | **n. +ve** | **n. Tested** | **n. +ve** | **n. Tested** | **n. +ve** |
| Darjeeling | 54 | 0 | 271 | 0 | 544 | 0 | 869 | 0 |
| East Champaran | 991 | 3 | 704 | 1 | 936 | 0 | 2361 | 4 |
| Godda | 1056 | 0 | 723 | 0 | 892 | 0 | 2631 | 0 |
| Gopalganj | 1080 | 0 | 702 | 0 | 456 | 0 | 2671 | 0 |
| Katihar | 893 | 0 | 207 | 0 | 586 | 0 | 1686 | 0 |
| Muzaffarpur | 699 | 0 | 817 | 0 | 484 | 0 | 2000 | 0 |
| Purnia | 505 | 0 | 253 | 0 | 632 | 0 | 1390 | 0 |
| Samastipur | 191 | 0 | 562 | 0 | 537 | 0 | 1290 | 0 |
| **Total** | 5469 | 3 | 4239 | 1 | 5067 | 0 | **14775** | 4 |

**Table 4. Case data for blocks associated with sentinel sites from KAMIS.**

| Sentinel Site | 2016 | | | 2017 | | | 2018 | | | 2019 | | |
|---|---|---|---|---|---|---|---|---|---|---|---|---|
| | Pop. | C. | Inc. | Pop. | C. | Inc. | Pop. | C. | Inc. | Pop. | C. | Inc. |
| Darjeeling—Phansidewa Block | 223341 | 0 | 0.00 | 227307 | 0 | 0.00 | 231344 | 9 | 0.39 | 235453 | 7 | 0.30 |
| East Champaran—Turkaulia Block | 205149 | 15 | 0.73 | 210500 | 10 | 0.48 | 215991 | 6 | 0.28 | 221625 | 4 | 0.18 |
| Godda—Poraiyahat Block | 206591 | 75 | 3.63 | 210639 | 60 | 2.85 | 214766 | 30 | 1.40 | 218975 | 19 | 0.87 |
| Gopalganj—Barauli Block | 291916 | 64 | 2.19 | 297872 | 44 | 1.48 | 303950 | 62 | 2.04 | 310151 | 51 | 1.64 |
| Katihar—Barari Block | 324329 | 18 | 0.55 | 332734 | 14 | 0.42 | 341358 | 11 | 0.32 | 350205 | 11 | 0.31 |
| Muzaffarpur -Minapur Block | 390877 | 42 | 1.07 | 401713 | 37 | 0.92 | 412850 | 27 | 0.65 | 424296 | 17 | 0.40 |
| Purnia—Dhamdaha Block | 320266 | 33 | 1.03 | 327122 | 43 | 1.31 | 334124 | 20 | 0.60 | 341277 | 10 | 0.29 |
| Samastipur—Warishnagar Block | 244214 | 9 | 0.37 | 250256 | 8 | 0.32 | 256447 | 7 | 0.27 | 262792 | 1 | 0.04 |
| **Total** | 2206683 | 256 | 1.16 | 2258145 | 216 | 0.96 | 2310831 | 172 | 0.74 | 2364773 | 120 | 0.51 |

Pop. Population calculated using Government of India 2011 Census [31] and population projected using the formula: Population Final = Population initial ((1+(growth rate/100))^(time in years))

stage development could be incorporated into the elimination efforts. When the vaccine programme failed, the elimination target year was revised to 2017 and then to 2020 [1,6].

The main strategies recommended for VL elimination are similar to those for malaria: (a) early case detection and complete treatment, (b) integrated vector management, (c) effective disease surveillance, (d) social mobilization and behavioural changes, and (e) operational research [13,15,33].

A range of different vector control interventions have been evaluated to control vectors of leishmaniasis. ITNs have proven useful for the control of *P. argentipes* in some communities [34,35], However, in India IRS has had the greater impact making it the preferred vector control tool in the Indian elimination campaign [8,9,16,36]. IRS for VL prevention has been used in India since 2005. To be effective IRS must be applied at a coverage and quality that should achieve the desired impact [37,38]. Resistance in the local vector to the insecticide used for IRS is also a potential threat to the programme success. DDT-based IRS from 2005–2014 failed to reduce transmission levels, in part due to coverage and quality of the IRS and high levels of resistance to DDT in *P. argentipes* [21]. In 2015 control efforts switched to a more effective insecticide, alpha-cypermethrin 5% wettable powder, to overcome resistance [22] and, stirrup pumps were replaced with compression pumps (Hudson X-pert Sprayer) to improve the quality of IRS delivery and increased training and monitoring efforts were implemented to improve coverage rates.

Data presented here show that there has been an obvious improvement in the IRS programme and VL elimination targets (<1:10,000 population) are close to being achieved in the region, prompting WHO to work with Bangladesh, India and Nepal to establish the data requirements for validation of elimination.

Routine entomological and IRS surveillance was embedded in the operational programme in 2016 to determine the impact of the IRS changes in the three VL endemic States; Bihar, Jharkhand and West Bengal (Fig 1).

The WHO target for IRS coverage is >80% of targeted structures being completely sprayed [23,32]. Self-reporting of IRS coverage by the spray teams suggests that >80% of households and >90% of cattle sheds were fully sprayed (Fig 2A). Initial independent household survey data suggests that this is an overestimate, with a lower complete spray coverage range (28% to 77%), although 80% is reached if partial house sprays are included. The partial spray can in part be accounted for by the homeowners only allowing certain rooms such as bedrooms to be

sprayed while not allowing storerooms or cooking areas to be sprayed. While the expected over reporting by the spray teams continues, there has been improvement in the overall trend towards higher complete spray coverage from 2016 to 2019, this is due to better engagement with communities on spray campaigns and the need to spray the complete structure. The largest improvement was seen in Godda, where actual coverage increased from 28% in 2016 to 75% in 2019.

For an increase in IRS coverage to have the desired impact, the correct dose of insecticide must be deposited onto the surface [38,39]. When DDT was applied using stirrup pumps the quality of IRS was well below that required [21,40,41]. Factors reducing the quality of the IRS included a sub-WHO specification formulation or the use of expired insecticide [40], rapid settling of the formulation in the pumps, compounded by the use of stirrup pumps. Spray operator performance has now been enhanced, with improved training, better quality assured formulations which form more even suspensions and the use of compressions pumps.

Quality assurance of the IRS remains an issue. The WHO recommends bioassays using susceptible insect vectors and/or sprayed filter paper analysis by HPLC. The former method is not possible, as no fully susceptible colony of *P. argentipes* exists, and collecting sufficient numbers of wild caught females to do routine bioassays is not feasible given current low densities of sand flies. Deploying the HPLC analysis of filter papers the average concentration of insecticide applied to surfaces was consistent from 2017 to 2019 (2017: 40.03mg/m$^2$, 2018: 41.64mg/m$^2$ and 37.92mg/m$^2$). This is 1.6 times above the target dose of 25mg/m$^2$. The reported level of over spraying is likely due to spray operators realising that the filter papers are being checked and ensuring that the papers are well sprayed. The actual amount of insecticide spayed per State is in line with the number of structures calculated to be sprayed at the target dose. An alternative method of measuring alpha-cypermethrin on walls, without the need for filter papers, is now in late-stage development [42] and should improve IRS quality assurance further.

*Phlebotomus argentipes* susceptibility to alpha-cypermethrin and other insecticides that might be used for IRS was monitored following the WHO insecticide resistance testing procedures [43]. As no diagnostic dose for resistance detection has been determined for sand flies, the WHO diagnostic dosages for malaria vectors were used as a surrogate.

The DDT *Anopheles* diagnostic dose works as a good surrogate. DDT resistance has increased from 2016–2019 (66.67% mortality in 2016 to 39.90% in 2019 (Chi$^2$ P<0.01), which is higher than when DTT was used for IRS [21]. This suggests that DDT resistance is still being selected for in Indian sand flies. There are two potential sources of selection. DDT is an extremely stable insecticide which decays slowly over many years. It is possible that the sublethal doses of DDT remaining on walls in areas sprayed for almost a decade with DDT are still exerting a selection pressure. Alpha-cypermethrin may also select directly for DDT resistance. DDT and pyrethroids (alpha-cypermethrin) both target the *para* volage-gated sodium channel in the insect nervous system [44]. Mutations in this channel gene, known as *kdr*, (knockdown resistance) are found at high frequencies in *P. argentipes* populations in Bihar [45].

While the *kdr* gene strongly predicts DDT resistance in *P. argentipes*, our bioassays with alpha-cypermethrin suggest that resistance conferred to this insecticide by *kdr* is likely to be low. IRS with pyrethroids for VL control has been used in Bangladesh with deltamethrin since 2012 [39] and Nepal has used alpha-cypermethrin and lambda-cyhalothrin on a rotation since 1992 [46] with no evidence of resistance selection.

As the diagnostic dosages for Anopheles insecticide resistance testing procedures are likely to be higher than those for sand flies; ideally the diagnostic concentration for monitoring sand flies should be ascertained to improve resistance monitoring and inform insecticide selection.

Currently alpha-cypermethrin is an effective insecticide for IRS control of *P. argentipes* in India. However, if IRS post-COVID-19 needs to be maintained for several more years, a proactive approach to insecticide resistance management [47] rotating through different classes of insecticide for IRS should be adopted.

The reduction in peak abundance of *P. argentipes* ranged from 81% to 86% per sentinel site over the study period, that exceeds the 67% reduction that models suggest is required for reaching the VL elimination target [17] (Fig 4). The declining trend in numbers of *P. argentipes* being caught suggest that the prolonged and extensive vector control programme has had a positive impact, lowering the abundance of vectors and reducing the transmission potential. A decline in *P. argentipes* has occurred in both the IRS and non-IRS villages. This may be explained by the close association of IRS and non-IRS villages in blocks and the potential spill over effect of the IRS control programme. As this was not a cluster randomised trial it is not possible here to compare the two village types directly. However, there is a need to ensure that the trends are not linked to long term sand fly population changes due to climatic variables [7].

Across all three states between 2017 and 2019 we detected only 0.03% of *P. argentipes* infected with *L. donovani* in IRS villages. This is significantly lower than observations in previous studies where *P. argentipes* infected with *L. donovani* ranged from 0.85% to 32% [48–52]. This low level of infection detected is due to both the impact of the case detection and treatment reducing the human reservoir of *L. donovani* [53] and the impact of IRS that will reduce the abundance and age of the vector [54,55]. The low abundance and low level of infection of *P. argentipes* suggests that active transmission in the region is now low, which is evidenced in the reduction in cases (Table 4).

The overall improvement in coverage and quality of IRS is associated with a reduction in the abundance of *P. argentipes* and the percentage that are infected with *L. donovani*. Combined with improved VL case detection and management the overall impact has been a reduction in disease transmission, which has allowed India to approach the VL elimination target of 1 in 10,000 VL cases at the block level [24]. Annual cases in India have gone from 32,803 in 2005, peaking at around 44,000 in 2007 and then declining by 90% to 3,128 cases in 2019 [56].

The biannual rounds of IRS are timed to coincide with optimum sand fly abundance patterns. From 2005 to 2014 this was often an issue with spraying starting late. Efforts have been increased in recent years to improve the timing of the IRS rounds. However, the second round of IRS in 2018 did not occur, due to operational issues. At this critical stage of approaching VL elimination targets, it is important that entire rounds of IRS are not missed as this could allow for a resurgence of *P. argentipes* and increase the selection pressure for insecticide resistance as the residues on the wall from the first-round diminish. An increase in the numbers of VL cases may then be triggered. The COVID-19 pandemic in 2020 delayed IRS activities, however, two rounds were completed in Bihar, Jharkhand and Utter Pradesh, while only one was completed in West Bengal. As India emerges from the COVID-19 pandemic, the impact of this on VL elimination efforts will need to be assessed.

## Conclusion

Due to the improved case detection, treatment and vector control, transmission of *L. donovani* by *P. argentipes* in India is currently low, which is evidenced in the reduction in cases. This success suggests that it is time for the programme to orientate in line with the WHO VL elimination guidelines [13,15] to the consolidation phase. In this phase the total coverage by spraying may no longer be required. At the same time there is a need for enhanced surveillance to detect increases in VL incidence or changes in the sand fly population so that any potential disease resurgence will elicit a rapid response.

## Supporting information

**S1 Table. Location of sentinel site villages and dates of indoor residual spraying.**
(DOCX)

**S2 Table. Indoor residual spraying by district that the sentinel sites are located as reported by spray operators.**
(DOCX)

**S3 Table. Indoor residual spray coverage data from household surveys.**
(DOCX)

## Acknowledgments

We thank the National Vector Borne Disease Control Programme for facilitating this work and the villages in which we have had the pleasure to work in.

## Author Contributions

**Conceptualization:** Sridhar Srikantiah, Michael Coleman.

**Data curation:** Rinki Deb, Rudra Pratap Singh, Arti Manorama Barwa, Debanjan Patra, Chandrima Das, Indranil Sukla, Ashish Kumar Srivastava, Shilpa Raj, Swikruti Mishra, Madhuri Swain, Swapna Mondal, Udita Mandal, Anna Trett, Gala Garrod, Laura McKenzie, Asgar Ali, Naresh K. Gill.

**Formal analysis:** Rinki Deb, Prabhas Kumar Mishra, Lisa Hitchins, Emma Reid, Laura McKenzie, Asgar Ali, Michelle C. Stanton, Michael Coleman.

**Funding acquisition:** Geraldine M. Foster, Janet Hemingway, Sridhar Srikantiah, Michael Coleman.

**Investigation:** Rinki Deb, Lisa Hitchins, Emma Reid, Arti Manorama Barwa, Debanjan Patra, Chandrima Das, Indranil Sukla, Ashish Kumar Srivastava, Shilpa Raj, Swikruti Mishra, Madhuri Swain, Swapna Mondal, Udita Mandal, Geraldine M. Foster, Gala Garrod, Laura McKenzie, Asgar Ali, Michael Coleman.

**Methodology:** Rinki Deb, Prabhas Kumar Mishra, Lisa Hitchins, Emma Reid, Geraldine M. Foster, Anna Trett, Gala Garrod, Laura McKenzie, Michelle C. Stanton, Sridhar Srikantiah, Michael Coleman.

**Project administration:** Rinki Deb, Rudra Pratap Singh, Prabhas Kumar Mishra, Anna Trett, Karthick Morchan, Indrajit Chaudhuri, Janet Hemingway, Sridhar Srikantiah, Michael Coleman.

**Resources:** Rinki Deb, Rudra Pratap Singh, Prabhas Kumar Mishra, Karthick Morchan, Indrajit Chaudhuri, Nupur Roy, Naresh K. Gill, Chandramani Singh, Neeraj Agarwal, Sadhana Sharma, Sridhar Srikantiah, Michael Coleman.

**Software:** Anna Trett, Gala Garrod.

**Supervision:** Rinki Deb, Rudra Pratap Singh, Prabhas Kumar Mishra, Arti Manorama Barwa, Debanjan Patra, Chandrima Das, Indranil Sukla, Ashish Kumar Srivastava, Shilpa Raj, Swikruti Mishra, Madhuri Swain, Swapna Mondal, Udita Mandal, Geraldine M. Foster, Asgar Ali, Karthick Morchan, Indrajit Chaudhuri, Chandramani Singh, Neeraj Agarwal, Sadhana Sharma, Janet Hemingway, Michael Coleman.

**Validation:** Rudra Pratap Singh, Emma Reid, Laura McKenzie.

**Writing – original draft:** Rinki Deb, Michelle C. Stanton, Janet Hemingway, Michael Coleman.

**Writing – review & editing:** Rinki Deb, Rudra Pratap Singh, Prabhas Kumar Mishra, Lisa Hitchins, Geraldine M. Foster, Anna Trett, Gala Garrod, Naresh K. Gill, Chandramani Singh, Neeraj Agarwal, Sadhana Sharma, Michelle C. Stanton, Janet Hemingway, Sridhar Srikantiah, Michael Coleman.

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
