## [Decision Letter · Decision Letter 0]

27 Mar 2021

Dear Dr Coleman,

Thank you very much for submitting your manuscript "Title - Impact of IRS: Four-years of entomological surveillance of the Indian Visceral Leishmaniases elimination programme." for consideration at PLOS Neglected Tropical Diseases. As with all papers reviewed by the journal, your manuscript was reviewed by members of the editorial board and by several independent reviewers. In light of the reviews (below this email), we would like to invite the resubmission of a significantly-revised version that takes into account the reviewers' comments. 

We cannot make any decision about publication until we have seen the revised manuscript and your response to the reviewers' comments. Your revised manuscript is also likely to be sent to reviewers for further evaluation.

Sincerely,

Olaf Horstick, FFPH(UK)

Associate Editor

Eric Dumonteil

Deputy Editor

Reviewer's Responses to Questions

**Key Review Criteria Required for Acceptance?**

**Methods**

-Are the objectives of the study clearly articulated with a clear testable hypothesis stated?

-Is the study design appropriate to address the stated objectives?

-Is the population clearly described and appropriate for the hypothesis being tested?

-Is the sample size sufficient to ensure adequate power to address the hypothesis being tested?

-Were correct statistical analysis used to support conclusions?

-Are there concerns about ethical or regulatory requirements being met?

Reviewer #1: - It should be “confidence level” when the authors stated the following at the beginning of page-4 of the manuscript: “Eight hundred households were targeted for each spray round in

- each of these districts assuming a 95% confidence interval, 5% absolute precision and 50% expected coverage based on the most conservative measure..” Did the authors also considered design effect, as they were dealing with clusters?

Reviewer #2: It is a descriptive study and no hypothesis was aimed to be tested.

Reviewer #3: Objectives articulated, Study design appropriate, population clearly described, sample size sufficient, correctly analysed and ethically cleared. Authors must clarify in which lab HPLC was performed. There is a defect that they had not send few samples to WHO recommended lab for cross check.

**Results**

-Does the analysis presented match the analysis plan?

-Are the results clearly and completely presented?

-Are the figures (Tables, Images) of sufficient quality for clarity?

Reviewer #1: Results are well analyzed and presented.

Reviewer #2: Yes, study results matched with the analysis plan. Results are presented adequately. However, discussion section needs better interpretation of the study results.

Reviewer #3: Analysis match and results clearly presented.

Number of Table numbered incorrectly, figures have some defects. Correction of these defects have been suggested.

**Conclusions**

-Are the conclusions supported by the data presented?

-Are the limitations of analysis clearly described?

-Do the authors discuss how these data can be helpful to advance our understanding of the topic under study?

-Is public health relevance addressed?

Reviewer #1: - The following statement at the bottom of page-9 is not correct.

“While a range of different vector control interventions for P. argentipes have been tested the only one to show success to date is IRS, limiting the options for vector control to a single intervention.” Other vector control measures, for example ITN, also work effectively in community settings, as shown in different studies.”

Reviewer #2: Study limitation is missing. Study data found decline in LD infection rate among sand fly against historical data, but the study did not have its own baseline information regarding this. Therefore it is hard to accept that the decline of LD infection among SF was due to IRS. Other possible factors must be discussed. The study results failed to demonstrate that improved IRS activities in the study areas succeed in drooping SF densities as the differences of SF densities between IRS and non-IRS areas were insignificant. Therefore, does the national program need to continue IRS which is a big burden for the Government like India in relation its cost, logistics, HR and infrastructure?

Reviewer #3: There is no separate column for conclusion. Limitations have not been described.

This publication will help scientists working in this field. This will also help policy makers of India, Nepal and Bangladesh to check their policy to eliminate Kala-azar.

**Editorial and Data Presentation Modifications?**

Reviewer #1: (No Response)

Reviewer #2: The need of IRS must be justified with adequate data, otherwise it has to be discouraged in the light of the study results.

Reviewer #3: (No Response)

**Summary and General Comments**

Reviewer #1: - One of the key elimination strategies was “integrated vector management and vector surveillance”. It did not focus on IRS only, as said in the Abstract and Introduction of the manuscript. 

- It would be better if the author says South Asia, instead of Indian subcontinent. It is no longer like that after the British left the region in 1947. They cleverly divided the region before leaving their beloved colony. 

- The elimination target is not described correctly. Please have a look at the referenced resources. Although, these are clearly presented under Discussion. 

- The elimination strategies mentioned are also incomplete and wrongly put vector control first. 

- The authors should have made it clear that DDT is (was) used in India only, and has been banned from using for vector control in other South Asian countries like Bangladesh. The authors cited reference (16) in the Introduction and mentioned about the usefulness of DDT. But in that study, only India used DDT. Neither Bangladesh nor Nepal study sites used DDT as intervention. The authors should have read the actual article before citing it in their manuscript. Or they might have overlooked that willingly, which is unscientific.

Reviewer #2: This is an observational study on IRS in relation to the National VL elimination program in India. The study had 8 sites for data collection. In India more than 600 blocks are endemic for VL. So inclusion of 8 sites represents only 1.6% of the all VL endemic areas which is insufficient regarding sample size calculation. Over sprayed filter papers with IRS for monitoring demonstrated that those were unblinded for spray men making them unacceptable for validation of spray quality. IRS coverage among IRS areas was consistently low than that recommended by the WHO. Further lack of differences in SF densities between IRS and non-IRS areas makes this intervention questionable for its further continuation. Authors shall highlights these findings in the discussion and make adequate recommendation.

Reviewer #3: These are the general suggestion/modifications/ questions to be addressed by authors to improve the paper.

Authors Summary:

Last line: VL elimination target of 1 case per 10,000 at the sub-district level.

Suggestion: India has no structure like sub-district; sub district level is target for Bangladesh. It should be less than 1 case per 10,000 population at…….. level.

Method

Each site had at least 1 new VL case per 10,000 capita per year at sub-district (block) level.

Q a: What do you mean by 10,000 capita per year?

Q b: What do you mean by sub-district level? Sub-districts are in Bangladesh. Please check tripartite agreement done in 2005 between Bangladesh, India, and Nepal.

Quality assurance:

Your line: and stored at -4 0C until analysis following IRS activities.

Suggestion: You must have noticed that impregnated papers were to be stored at room temperature, similarly to keep at -4 oc is not recommended. Hence remove these words.

You: HPLC was performed…..

Question: In which lab it was performed? Name the lab. Whether some sample was sent to another lab to cross check the results? You must have read that different labs had different results. Hence, few samples must be cross checked from WHO recommended lab.

Insecticide Susceptibility Assays

You: and Abbott’s formula applied to correct for control mortality

Suggestion: a. For readers provide either Abbott’s formula or the reference.

b. Abbott’s formula is not used to correct control mortality but it is used to correct test mortality if control mortality is within 20%. Correct the text accordingly.

Phlebotomus argentipes abundance.

You: by morphological criteria from established taxonomic keys (28).

Suggestion: Correct the reference in reference section.

Results 

Indoor residual spraying data

You:. …spray operators reporting over 80% coverage of households and cattle sheds covered with alpha-cypermethrin at 0.25g/m2 IRS. 

Suggestion: correct the dose according to methodology section. 

IRS coverage as measured from the survey data:

You: In Bihar, consistently low-level of complete coverage over the four years was observed in Samastipur (49-61%).

Suggestion: What were the causes? Discusses them in discussion section. Provide your suggestions to overcome the problem.

Quality assurance:

A total of 642 houses that received IRS across the eight sites were included in the QA surveys. Four filter papers per wall were affixed in the bedroom prior to IRS and recovered afterwards. A total of 3,050 Whatman filter papers were analysed by HPLC over three years (2017-2019). In 2017, all 2,140 filter papers collected from field surveys were analysed. In subsequent years a random sample of 20% of houses from each sentinel site were analysed.

Question: You had taken 642 houses and 4 filter papers per room (642 x 4= 2568). Then from where you have collected 3050 samples for HPLC)? If your second line is considered 20% of 642 houses is 128 x 4 walls = 512. Hence, total sample will be 2568+512= 3080. Justify and rewrite this paragraph.

You: Of the 2,944 filter papers tested from 2017 t002019….

Question: Now from where this number arrived? Which of the samples were discarded? 

You: In East Champaran low quality spraying increased annually from 48.3% in round 1 of 2017 to 70% in round 2 of 2019. Suggestion: Re-frame the sentence to clear the meaning (low quality increased or quality improved?).

You: No resistance was detected to the other insecticides tested, Table 2 (21).

Suggestion: a. Where is your Table 1 (in the text Table 1 is missing).

b. In results section your findings should be explained, You should not place any reference like 21.

You: in IRS villages, and here we report a reduction to 0.75 P. argentipes/trap/night for the same time period in 2019. (21), in IRS villages, and here we report a reduction to 0.75 P. argentipes/trap/night for the same time period in 2019.

Suggestion: Duplication of line.

Your Table 2: During 2016 and 2019, why mortality in low concentration is high for Alpha cypermethrin? Discuss.

Figure 2. Percentage of household structures that were completely sprayed in each IRS round as reported by; A spray operators in all 8 sites, and B sentinel sites surveys. R1= Round one R2=Round two. Solid colors represent complete spray and lined colors partial spray. 

Question a: you have 8 sentinel sites. 1. One bar is missing in all the cases except 2018 R1.

b. In bar there are colors but you have not correlated the colors to different sites.

Fig 3: has been labelled as Fig 2.

Suggestion: This fig. is related to dose. Colors are indicated as spray/target /spray. Correct them. 

Figure 4: Aggregated P. argentipes abundance 

Question: Why the abundance of P. argentipes is high in IRS in comparison to non-IRS in July 2017? Discuss.

PLOS authors have the option to publish the peer review history of their article (what does this mean?). If published, this will include your full peer review and any attached files.

Reviewer #1: No

Reviewer #2: No

Reviewer #3: Yes: Prof. Dr Murari Lal Das
---

## [Decision Letter · Decision Letter 1]

28 Jun 2021

Dear Dr Coleman,

Thank you very much for submitting your manuscript "Title - Impact of IRS: Four-years of entomological surveillance of the Indian Visceral Leishmaniases elimination programme." for consideration at PLOS Neglected Tropical Diseases. As with all papers reviewed by the journal, your manuscript was reviewed by members of the editorial board and by several independent reviewers. The reviewers appreciated the attention to an important topic. Based on the reviews, we are likely to accept this manuscript for publication, providing that you modify the manuscript according to the review recommendations. 

Some minor edits suggested by one reviewer

Sincerely,

Olaf Horstick, FFPH(UK)

Associate Editor

Eric Dumonteil

Deputy Editor

Some minor edits suggested by one reviewer

Reviewer's Responses to Questions

**Key Review Criteria Required for Acceptance?**

**Methods**

-Are the objectives of the study clearly articulated with a clear testable hypothesis stated?

-Is the study design appropriate to address the stated objectives?

-Is the population clearly described and appropriate for the hypothesis being tested?

-Is the sample size sufficient to ensure adequate power to address the hypothesis being tested?

-Were correct statistical analysis used to support conclusions?

-Are there concerns about ethical or regulatory requirements being met?

Reviewer #1: The authors have correctly addressed the comments made on the Methods section of the first submission of the manuscript. Thank you.

Reviewer #3: All right

**Results**

-Does the analysis presented match the analysis plan?

-Are the results clearly and completely presented?

-Are the figures (Tables, Images) of sufficient quality for clarity?

Reviewer #1: (No Response)

Reviewer #3: All right

**Conclusions**

-Are the conclusions supported by the data presented?

-Are the limitations of analysis clearly described?

-Do the authors discuss how these data can be helpful to advance our understanding of the topic under study?

-Is public health relevance addressed?

Reviewer #1: The authors have correctly addressed the comments made on the Discussion section of the first submission of the manuscript. Thank you.

Reviewer #3: All right

It needs editing by authors/editors (some words highlighted in the attachment).

**Editorial and Data Presentation Modifications?**

Reviewer #1: (No Response)

Reviewer #3: Some typing mistakes are still there, that might be corrected either by editors or authors, listed below.

1. Affiliation of Indrajit Chaudhuri is still missing.

2. Authors summary and case data from sentinel sites (under results): India is now on track to reach it’s target incidence for VL of less than 1/1000 people at the sub-district (block) level… and Other than Gopalganj, the threshold for VL elimination of an incidence less than 1/1000 people at the block level, was reached- respectively.

3. Reference 46. Division EaDC- should be corrected

4. Conclusion should be edited by authors/ editors (words highlighted in the attachment).

**Summary and General Comments**

Reviewer #1: I would like to thank the authors for carefully addressing all the comments that I made on the first submission of the manuscript. This is much appreciated. I consider this manuscript as an important addition to the knowledge base on kala-azar vector control programs.

Reviewer #3: All right

PLOS authors have the option to publish the peer review history of their article (what does this mean?). If published, this will include your full peer review and any attached files.

Reviewer #1: No

Reviewer #3: Yes: Prof. Dr Murari Lal Das

Figure Files:

Data Requirements:

Reproducibility:

References

---

## [Editor Report · Decision Letter 2]

1 Jul 2021

Dear Dr Coleman,

We are pleased to inform you that your manuscript 'Title - Impact of IRS: Four-years of entomological surveillance of the Indian Visceral Leishmaniases elimination programme.' has been provisionally accepted for publication in PLOS Neglected Tropical Diseases.

Best regards,

Olaf Horstick, FFPH(UK)

Associate Editor

Eric Dumonteil

Deputy Editor

---

## [Editor Report · Acceptance letter]

30 Jul 2021

Dear Dr Coleman,

We are delighted to inform you that your manuscript, "Title - Impact of IRS: Four-years of entomological surveillance of the Indian Visceral Leishmaniases elimination programme," has been formally accepted for publication in PLOS Neglected Tropical Diseases.

Best regards,

Shaden Kamhawi

co-Editor-in-Chief

Paul Brindley

co-Editor-in-Chief
